# Morphological Alterations and Stress Protein Variations in Lung Biopsies Obtained from Autopsies of COVID-19 Subjects

**DOI:** 10.3390/cells10113136

**Published:** 2021-11-12

**Authors:** Rosario Barone, Antonella Marino Gammazza, Letizia Paladino, Alessandro Pitruzzella, Giulio Spinoso, Monica Salerno, Francesco Sessa, Cristoforo Pomara, Francesco Cappello, Francesca Rappa

**Affiliations:** 1Department of Biomedicine, Neurosciences and Advanced Diagnostics (BiND), Institute of Human Anatomy and Histology, University of Palermo, 90127 Palermo, Italy; rosario.barone@unipa.it (R.B.); antonella.marinogammazza@unipa.it (A.M.G.); letizia.paladino@unipa.it (L.P.); alessandro.pitruzzella@unipa.it (A.P.); giulio.spinoso@community.unipa.it (G.S.); francesca.rappa@unipa.it (F.R.); 2Euro-Mediterranean Institutes of Science and Technology (IEMEST), 90139 Palermo, Italy; 3Consortium of Caltanissetta, 93100 Caltanissetta, Italy; 4Department of Medical, Surgical and Advanced Technologies “G.F. Ingrassia”, University of Catania, 95121 Catania, Italy; monica.salerno@unict.it (M.S.); cristoforo.pomara@unict.it (C.P.); 5Department of Clinical and Experimental Medicine, University of Foggia, 71122 Foggia, Italy; francesco.sessa@unifg.it

**Keywords:** SARS-CoV-2, COVID-19, Hsp60, Hsp90, endothelium, inflammation

## Abstract

Molecular chaperones, many of which are heat shock proteins, play a role in cell stress response and regulate the immune system in various ways, such as in inflammatory/autoimmune reactions. It would be interesting to study the involvement of these molecules in the damage done to COVID-19-infected lungs. In our study, we performed a histological analysis and an immunomorphological evaluation on lung samples from subjects who succumbed to COVID-19 and subjects who died from other causes. We also assessed Hsp60 and Hsp90 distribution in lung samples to determine their location and post-translational modifications. We found histological alterations that could be considered pathognomonic for COVID-19-related lung disease. Hsp60 and Hsp90 immunopositivity was significantly higher in the COVID-19 group compared to the controls, and immunolocalization was in the plasma membrane of the endothelial cells in COVID-19 subjects. The colocalization ratios for Hsp60/3-nitrotyrosine and Hsp60/acetylate-lisine were significantly increased in the COVID-19 group compared to the control group, similar to the colocalization ratio for Hsp90/acetylate-lisine. The histological and immunohistochemical findings led us to hypothesize that Hsp60 and Hsp90 might have a role in the onset of the thromboembolic phenomena that lead to death in a limited number of subjects affected by COVID-19. Further studies on a larger number of samples obtained from autopsies would allow to confirm these data as well as discover new biomarkers useful in the battle against this disease.

## 1. Introduction

COVID-19, the disease induced by SARS-CoV-2 (a new coronavirus discovered for the first time in China in 2019), has caused a pandemic worldwide, starting about one year ago [1]. Its severity is due to two factors: the high index of contagiousness and the fact that it can lead to death relatively quickly in a small cohort of subjects.

Essentially, the disease consists of four stages [2]: (1) Asymptomatic. Here, the subject does not know that they are infected with the virus but can still transmit it. (2) Symptomatic with upper airway symptoms. Here, the disease presents like a common cold or influenza, and only positivity to the swab reveals that the subject is infected with SARS-CoV-2. (3) Symptomatic with pulmonary involvement. In this phase, the subject’s immune system has lost control of the disease. There is a hyperimmune response, probably also accompanied by an autoimmune response against small blood vessels. The lung, which is very rich in capillaries, is among the first organs to be affected, and its microvascular destruction determines the most significant symptomatology, i.e., a sharp drop in oxygen saturation of the blood. (4) Systemic. In this phase, the destruction of the small vessels is widespread in many organs throughout the body and the subject, especially if they are predisposed due to other concomitant pathologies, can experience multiorgan failure induced (or accompanied) by disseminated intravascular coagulation (DIC), with a high chance of not surviving.

A long list of predisposing factors for the worsening of the clinical picture has been published in various scientific articles. However, in our opinion all these factors are attributable to two basic causes: hypertension and diabetes [3,4,5,6,7]. In fact, these two conditions not only represent the primum movens for the development of more severe diseases (such as myocardial infarction, heart failure, metabolic syndrome, stroke, etc.), many of which become symptomatic only several years from their onset, but they are also considered key stress factors affecting endothelial cells. More specifically, hypertension causes physical stress to these cells, while diabetes induces chemical stress [8,9,10,11].

Endothelial cells react to stress in various ways. Among these is the induction of post-translational changes in proteins that are normally intracellular, which, as a result of these changes, localize at the level of the plasma membrane due to molecular changes that are not yet well understood [12,13,14]. Putatively, if these proteins expose epitopes on the cell surface that are targeted as non-self by the immune system, the subject can develop an immune reaction against these cells, resulting in endothelialitis, which can be further complicated by the formation of thrombus and emboli.

Among the endothelial proteins that can undergo these changes are the two molecular chaperones Hsp60 and Hsp90 [15,16]. Normally, in endothelial cells and beyond, Hsp60 is found mainly inside the mitochondria, while Hsp90 can also be found constitutively at the membrane level, other than in the cytosol. For Hsp60, stress episodes can determine post-translational changes and its dislocation at the membrane level, with the triggering of inflammatory/autoimmune phenomena affecting these cells [17,18,19].

When the first autopsies conducted on COVID-19 subjects in Italy revealed that the damage to the lungs was mainly of a thromboembolic nature and associated with signs of endothelialitis, it immediately became clear that this type of damage could be induced by an autoimmunity response triggered by a phenomenon known as “molecular mimicry” [20,21]. The hypothesis was that some viral proteins could have a common epitope with human proteins, and immune activation against the virus, in some predisposed subjects, could therefore cause an autoimmune reaction against endothelial cells.

A screening of all human proteins compared to all COVID-19 proteins was successively performed, and results refined with the use of software revealed the immunogenicity of the epitopes found [22,23,24,25]. The most significant findings were recently published, and among the human proteins suspected of triggering the phenomenon of molecular mimicry, there were also some molecular chaperones, including Hsp60 and Hsp90.

Autopsy remains the gold standard to determine why and how a death happens, and it may also provide useful clinical and epidemiologic insights. Selective approaches to postmortem diagnosis, such as limited postmortem sampling over full autopsy, can also be useful in the control of disease outbreaks and provide valuable knowledge for appropriate control measure management. Full autopsies on patients who died with suspected or confirmed COVID-19 infection could be used to investigate several biomarkers, which can allow us to gather more information on this disease, its prevention, and, above all, therapeutic approaches [21,26]. For this study, we decided to carry out a morphological and molecular study on lung samples obtained from autopsies of subjects who succumbed to severe forms of COVID-19, trying to uncover histological evidence on the involvement of Hsp60 and Hsp90 in the onset of endothelialitis and in the thromboembolic events. The results obtained are reported below.

## 2. Materials and Methods

### 2.1. Sample Collection

A total of 12 complete autopsies (five males and seven females) were split into two groups (Table 1): a COVID-19 group composed of six subjects (four females and two males) who died from COVID-19 (mean age 65 ± 12.8) and a control group composed of six subjects (three females and three males) who died from other causes (mean age 62 ± 10.3). Nasopharyngeal swabs collected from the COVID-19 subjects prior to death had tested positive in the COVID-19 rRT-PCR assay. Infection control strategies for safe management of autopsies were applied [27]. All autopsies were conducted according to the Letulle technique, consisting of the extraction en masse of the oral, cervical, thoracic, and abdominal viscera, with a final resection from the surrounding structures. Removing all the viscera using the Letulle technique reduces the potential for environmental contamination [28]. All organs were fixed in formalin. All procedures performed in the study were approved by the Scientific Committee of the Department of Medical and Surgical Sciences and Advanced Technologies “G.F. Ingrassia”, University of Catania, Italy (record n. 21/2020), and performed in accordance with the 1964 Helsinki Declaration and its later amendments or comparable ethical standards. The study was conducted according to the Italian Legislative Decree No. 81/2008 regulating health and safety at workplaces. The director of the San Marco Hospital, Catania, Italy, authorized the use of anonymous data according to current Italian and European regulations. No informed consent is required to use data related to deceased subjects in cases where such data is indispensable and relevant for scientific and research purposes.

### 2.2. Histological Analysis

For histological analysis, lung samples from six COVID-19 subjects (COVID-19 group) and six control subjects (control group) were selected. All samples were fixed in formalin and embedded in paraffin. Sections of 4–5 µm thickness were prepared from all paraffin blocks using a cutting microtome. These sections, placed on slides, were dewaxed in xylene for 10 min and rehydrated by sequential immersion in a descending scale of alcohols and transitioned in water for five minutes. The slides were then stained with hematoxylin and eosin, mounted with coverslips, and finally observed with an optical microscope (Microscope Axioscope 5/7 KMAT, Carl Zeiss, Milan, Italy) connected to a digital camera (Microscopy Camera Axiocam 208 color, Carl Zeiss, Milan, Italy).

### 2.3. Immunohistochemistry

Immunohistochemical (IHC) investigations were carried out on 5 µm thick sections obtained from paraffin blocks using a cutting microtome. The IHC reactions were performed using the automated IHC system of the Biotechnology Laboratory of the Euro-Mediterranean Institute of Sciences and Technologies (IEMEST) (IntelliPath Flx, Biocare Medical, distributed by Bio-Optica, Milan, Italy). The primary antibodies used are indicated in Table 2. At the end of the immunostaining cycle, the slides were prepared for observation with coverslips using a permanent mounting medium (VectaMount, Vector, H-5000, Burlingame, CA, USA). Observation of the sections was performed with an optical microscope (Microscope Axioscope 5/7 KMAT, Carl Zeiss, Milan, Italy) connected to a digital camera (Microscopy Camera Axiocam 208 color, Carl Zeiss, Milan, Italy). Two independent pathologists (F.C. and F.R.) examined the specimens on two separate occasions and performed a semiquantitative analysis to determine the percentage of cells positive for Hsp60 and Hsp90. The percentage of immunopositivity was evaluated in a high-power field (HPF) at 400× magnification and repeated for 10 HPF. The average of the percentages of all immunosemiquantifications performed in each case for the two groups was considered as a conclusive result, and this value was used for the statistical analysis.

### 2.4. Immunofluorescence

Formalin-fixed lung samples (n = 6 per group) were embedded in paraffin and cut into 5 µm sections. Immunofluorescence was performed as reported previously [29]. Paraffin sections were deparaffinized and rehydrated, and antigen retrieval was performed using citrate buffer pH 6 at 70 °C for 10 min. After blocking with bovine serum albumin (BSA, Sigma-Aldrich, St. Louis, MO, USA) for 30 min, the slides were incubated with the primary antibodies overnight at 4 °C. The primary antibodies used are indicated in Table 2. The following day, the samples were incubated with a species-specific fluorescent secondary antibody conjugated with Atto 488 or Atto 647 (dilution 1:100, Sigma-Aldrich, St. Louis, MO, USA). The nuclei were counterstained with DAPI dihydrochloride 32670 (dilution 1:1000, Sigma-Aldrich, St. Louis, MO, USA), and slides were covered with drops of PBS and mounted with coverslips. The images were captured using a Leica Confocal Microscope TCS SP8 (Leica Microsystems, Wetzlar, Germany). The staining intensity, for Hsp60 and Hsp90 of each sample, was expressed as the mean pixel intensity (PI) normalized to the cross-sectional area (CSA) using the Leica Application Suite Advanced Fluorescence software. Colocalization ratio (%) analysis was performed to test colocalization for Hsp60/CD34 and Hsp90/CD34 proteins.

### 2.5. Statistical Analyses

All data were collected and analyzed for statistical significance. Student’s *t*-test was used to identify statistically significant differences between the two groups. All statistical analyses were performed using the GraphPad PrismTM 4.0 software (GraphPad Software Inc., San Diego, CA, USA). All data are presented as the mean ± SD, and the threshold of statistical significance was set at *p* < 0.05.

## 3. Results

### 3.1. Histological Analysis

Histological analysis was performed on both COVID-19 and control lung samples stained with hematoxylin and eosin. In the COVID-19 group, the parenchyma architecture was significantly modified by a marked congestion with edema and microthrombi of the small vessels (Figure 1a) and hemorrhages in the interstitium and alveolar space (Figure 1b), in which histiocytes and inflammatory cells were found (Figure 1c). The thickness between capillaries and alveolus was significantly increased. The alveolar damage observed was caused by the destruction of the alveolar wall itself and the detachment of the alveolar lining, with desquamation of pneumocytes within the alveolar space (Figure 1d). Multiple type II pneumocytes were often seen forming aggregates similar in appearance to multinucleated giant cells within the alveolar space (Figure 1e). Abnormal cells with large, irregular, and monstrous nuclei were present in the lung interstitium (Figure 1f). In the control group, the parenchyma had a normal appearance with intact alveolar walls and empty alveolar spaces (Figure 1g,h).

### 3.2. Immunohistochemistry

The immunohistochemical evaluation of CKAE1AE3 and CK7 showed hyperplasia of type II pneumocytes in some areas of the COVID-19 lung parenchyma, with the pneumocytes appearing considerably enlarged with irregular and pleomorphic nuclei, sometimes presenting binucleation and cytopathic changes (Figure 1i,j). Some type II pneumocytes formed aggregates with a morphological appearance of multinucleated giant cells, probably due to hyperplasia and viral infection. These aggregates of cells were positive to CKAE1AE3 and CK7 (Figure 1l,m). Hyaline membranes originating from the destruction of the epithelial lining were marked by cytokeratin and did not have a fibrous origin (Figure 1i,l). The control tissues presented a normal alveolar wall architecture (Figure 1o,p). The immunohistochemical experiments for CD34 carried on COVID-19 lung tissue showed vascular congestion and increased microvascular texture (Figure 1k,n). Furthermore, in some areas, the wall of the microvessels was damaged and discontinuous (Figure 1n). In the control lung tissue, the microvessels displayed a normal architecture and their walls were continuous (Figure 1q).

Immunohistochemical experiments were performed only in COVID-19 lung parenchyma for CD61 (megakaryocyte marker), CD68 (histiocytic marker), and Ki67 (proliferation index) to better characterize the cells with abnormal and irregular nuclei present in the lung interstitium. The immunostain showed that these cells were positive to CD61, a megakaryocytes marker (Figure 1r), and localized in the vascular and extravascular space, as observed in the immunostaining with CD34 (Figure 1s). These data indicate that these cells are most likely megakaryocytes present within the vessels. Furthermore, these cells were negative to CD68 and Ki67 (Figure 1t,u), so they are neither histiocytes nor proliferating.

The immunopositivity for Hsp60 was evaluated on epithelial cells of the alveolar wall in the COVID-19 and control groups (Figure 2a,c,e). The results obtained showed that Hsp60 immunopositivity was significantly higher in the COVID-19 group (84.8 ± 2.7) than in the control group (31.5 ± 2.5) (*p* < 0.0001) (Figure 2g). The positivity for Hsp60 observed in COVID-19 samples was cytoplasmic in the hyperplastic pneumocytes and their aggregates, with a granular and diffuse pattern (Figure 2a,c). The pneumocytes of the control group showed slight cytoplasmic immunopositivity with a granular pattern (Figure 2e), typical for mitochondrial positivity of this protein.

The immunohistochemical results of Hsp90 experiments showed immunopositivity located in the cytoplasm of epithelial cells in both groups (Figure 2b,d,f). The results obtained showed that Hsp90 immunopositivity was significantly higher in the COVID-19 group (80.0 ± 2.8) than in the control group (31.8 ± 3.0) (*p* < 0.0001) (Figure 2h). The immunolocalization of Hsp90 in COVID-19 lung samples was observed in the cytoplasm, with a diffuse pattern in the hyperplastic pneumocytes and their aggregates (Figure 2b,d). The pneumocytes of the control group showed slight cytoplasmic immunopositivity with a granular and diffuse pattern (Figure 2f). High positivity was found in inflammatory cells present in the COVID-19 group samples (Figure 2b,d).

### 3.3. Confocal Microscopy Analysis

The immunofluorescence evaluation for Hsp60 and Hsp90, performed with confocal microscopy, confirmed the results obtained from the immunohistochemical experiments (Figure 3). The staining intensity, for Hsp60 and Hsp90 in each sample, was expressed as the mean PI normalized to the CSA using the Leica Application Suite Advanced Fluorescence software. As shown in the histograms in Figure 3k,w, the tissue levels of Hsp60 and Hsp90 were significantly higher in the COVID-19 group than in the control group (*p* = 0.0256 and *p* = 0.0280 respectively).

To evaluate localization of Hsp60 and Hsp90 in the walls of the vessels, we performed a double immunofluorescence for Hsp60/CD34 and Hsp90/CD34 (Figure 3). The colocalization ratio between Hsp60 and CD34 was significantly increased in the COVID-19 group compared to the control group (*p* = 0.0192) (Figure 3l). The double immunofluorescence for Hsp90/CD34 showed no significant difference in the colocalization ratio (%) between the two groups (Figure 3x).

Post-translational modification (PTM) is a covalent change in an amino acid in a protein that can modify its properties and functions, for instance, folding, ligand binding, migration, interaction with other molecules, and other specific roles [30]. For this reason, double immunofluorescence experiments were performed to determine Hsp60 and Hsp90 levels of nitration, acetylation, and phosphorylation on specific residues. The colocalization ratio between Hsp60 and 3-nitrotyrosine (Figure 4a–e) was significantly increased in the COVID-19 group compared to the control (*p* = 0.0137) (Figure 4m). However, no difference was observed in double immunofluorescence for Hsp90/3-nitrotyrosine (Figure 4g–l) between the groups (Figure 4n). A significant increase in the colocalization ratio was observed in the COVID-19 group compared to the control group both for Hsp60/acetylate-lisine (Figure 5a–e) and Hsp90/acetylate-lisine (Figure 5g–l) (*p* = 0.0185 and *p* = 0.0282 respectively) (Figure 5m,n). Finally, no difference was observed in the colocalization ratio for Hsp60/phosphotyrosine and Hsp90/phosphotyrosine between the groups (Figure 6).

## 4. Discussion

The first relevant data that we present in this work is undoubtedly the identification and characterization of some cells that could become pathognomonic for pulmonary involvement diagnostics in COVID-19. In particular, (i) the presence of type II pneumocytes within the alveolar space, which are often seen forming aggregates similar in appearance to multinucleated giant cells; (ii) the presence of abnormal cells with large, irregular, and monstrous nuclei in the interstitium. The former is immunohistochemically positive to CKAE1AE3 and CK7, confirming their epithelial nature, while the latter are positive to CD61, a megakaryocyte marker, revealing that these cells are most likely megakaryocytes. While the presence of multinucleated type II pneumocytes may indicate the attempt of the epithelial lining of the alveoli to regenerate the acini that were destroyed by the hyper/autoimmune reaction, the presence of megakaryocytes may be an epiphenomenon caused by the increased demand for platelets during the DIC. These findings add some histopathological clues to what has been described so far [31,32,33,34].

Another data of some importance to us is the finding that both Hsp60 and Hsp90 are increased in the endothelial cells of COVID-19 subjects compared to controls. In particular, this increase is more marked for Hsp60 which, more significantly than Hsp90, is localized in endothelial cells at the plasma membrane level, thus becoming potentially recognizable by the immune system. A possible molecular explanation for this dislocation at the membrane level may lie in the post-translational modifications, which we also found in our samples. These data, particularly those referred to Hsp60, are in agreement with what was postulated by Wick and colleagues, i.e., that the presence of Hsp60 in the plasma membrane of endothelial cells can trigger an inflammatory/autoimmune reaction [35,36,37], determining loss of endothelial cells of the vessels and, in turn, putatively triggering thrombosis in microvessels.

Hsp60 and Hsp90, as already described by our research group, have epitopes in common with some SARS-CoV-2 proteins [25]. In particular, the hexapeptide KDKKKK shared between SARS-CoV2 nucleoprotein and Hsp90 is part of the five experimentally validated epitopes of immunological relevance already catalogued in the Immune Epitope Database analysis resource (IEDB, https://www.iedb.org/, (accessed on 19 May 2021)) and correlates with the onset of Guillain–Barré syndrome [38]. Similarly, the hexapeptide EIPKEE shared between SARS-CoV2 Orf1ab polyprotein and Hsp60 is part of an experimentally validated autoimmune epitope catalogued in the IEDB and recognized by lymphomononuclear cells of multiple sclerosis (MS) patients [38,39]. As showed in Appendix A, the peptide EIPKEE is part of Hsp60 C-terminal residues, while the peptide KDKKKK of Hsp90 is included in the middle domain. However, these findings do not necessarily mean that there was an immune cross-reaction against Hsp60 and/or Hsp90 in the subjects included in the present study and that these proteins are invariably involved in all cases of death from COVID-19. In fact, we are well aware that the increase in Hsp60 and Hsp90 could be a consequence of the enormous cellular stress in the lungs of these subjects and can therefore be a secondary phenomenon. However, the existence of these common epitopes between viral proteins and these heat shock proteins in greater concentration and in ectopic locations (plasma membrane) in the endothelial cells of COVID-19 lung samples do not allow us to exclude their primary involvement and must lead to further studies on a larger scale to completely discard this hypothesis.

How could proving this hypothesis help in the fight against COVID-19? We have already stated that the transformation from a simple flu-like disease into a systemic disease characterized by hyper- and/or autoimmune activation cannot be predicted and that any such occurrence usually involves predisposed subjects, with diabetes and hypertension being the main risk factors for chronic and systemic stress at the endothelial level.

Finding markers (within the tissue and/or circulating) of predisposition can help clinicians intervene early in the cases most at risk. In fact, even though it seems clear by now that the transition from the “high respiratory” phase to the “pulmonary” phase is the most critical moment of the disease, in which it is necessary to intervene therapeutically with corticosteroids and heparin (the first to reduce the autoimmunity response attacking the vessels and the second to counter DIC) [40,41], a better understanding of the mechanisms of action of these drugs would help to choose the most precise and accurate dosage and timing.

Furthermore, knowledge, not only from a clinical point of view but also from a molecular point of view, on the risk factors predisposing subjects to the most severe complications of the disease would help protect the populations most at risk. Last but not least, identifying the proteins (viral and human) involved in the molecular mimicry phenomena could help produce increasingly safe and effective vaccines. In fact, is it possible to completely exclude the possibility that some of the fatal thromboembolic phenomena that occurred after the administration of a COVID-19 vaccine could have been caused by molecular mimicry between the SARS-CoV-2 spike protein and proteins present in the endothelial cells, at least in a very limited number of cases? Only in-depth tissue and molecular studies will definitively shed light on this too.

In conclusion, we wanted to, first of all, make a useful contribution to the collective effort to better understand COVID-19, with the aim of developing new studies that can lead to more comprehensive insight on the risk factors predisposing subjects to the onset of the most serious complications of SARS-CoV-2 infection. In particular, we hypothesize that the two stress proteins studied in this work, and Hsp60 in particular, may play an active role in the onset of the thromboembolic phenomena that has leads to death in a limited number of subjects affected by COVID-19. The observation of Hsp60 and Hsp90 modifications in the COVID-19 pathogenesis was possible thanks to the opportunity to study lung samples obtained during autopsies. Only access to a complete set of histological samples obtained through autopsy enables determination of the exact cause(s) of death, thereby optimizing clinical management and providing appropriate assistance to clinicians in pointing out a timely and effective treatment to reduce mortality. We hope that autopsy samples will continue to be used to clarify further aspects of the pathogenesis of this dramatic disease in the future.

## Figures and Tables

**Figure 1 cells-10-03136-f001:**
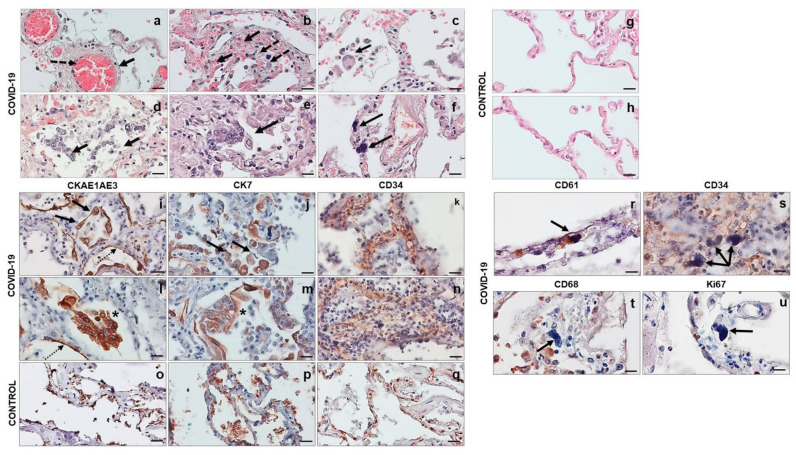
Histological staining of the lung parenchyma. (**a**–**h**) Hematoxylin and eosin staining of COVID-19 and control lungs. The parenchyma showed marked congestion with microthrombi of the small vessels ((**a**), arrow) and edema ((**a**,**b**), dotted arrows). There was visible hemorrhage in the interstitium and alveolar space ((**b**), arrows), in which histiocytes and inflammatory cells ((**c**), arrow) and desquamated hyperplastic pneumocytes ((**d**), arrows) were found. Aggregates of type II pneumocytes ((**e**), arrow) and cells with large, irregular, and monstrous nuclei ((**f**), arrows) were evident. The control parenchyma showed a normal appearance with intact alveolar wall and empty alveolar space (**g**,**h**). (**i**–**q**) Representative images of immunohistochemical reactions in COVID-19 and control lungs. (**i**,**j**) Hyperplasia of type II pneumocytes stained with CKAE1AE3 ((**i**), arrows) and CK7 ((**j**), arrows). (**l**,**m**) Aggregates of type II pneumocytes, positive to CKAE1AE ((**l**), showed with an asterisk in the figure) and CK7 ((**m**), asterisk). The dotted arrows indicate the hyaline membranes originating from the destruction of the epithelial lining and stained with CKAE1AE3 (**i**,**l**). (**k**,**n**) CD34 showed vascular congestion and increased microvascular texture. Normal microvascular architecture was present in control lung (**q**). (**a**–**q**) Magnification 400×, scale bar 20 µm. (**r**–**u**) Representative images of immunohistochemical reactions in COVID-19 lung. The cells with abnormal nuclei were positive to CD61 ((**r**), arrow); they were located inside the blood vessels ((**s**), arrows) and were negative for CD68 ((**t**), arrow) and Ki67 ((**u**), arrow). (**r**–**u**) Magnification 630×, scale bar 20 µm.

**Figure 2 cells-10-03136-f002:**
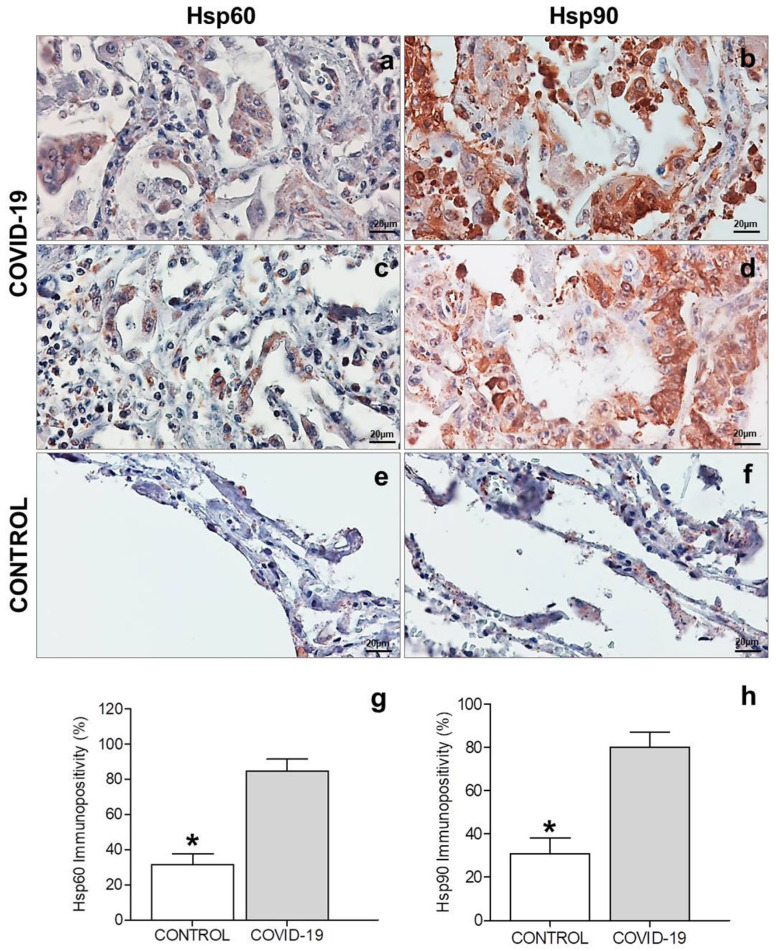
Immunohistochemistry for Hsp60 and Hsp90. (**a**,**c**,**e**) Representative images of immunohistochemical detection of Hsp60 in lung parenchyma of the COVID-19 and control groups. (**g**) Histogram showing the percentage of Hsp60-positive epithelial cells in the control and COVID-19 groups. (**b**,**d**,**f**) Representative images of immunohistochemical detection of Hsp90 in lung parenchyma of the COVID-19 and control groups. (**h**) Histogram showing the percentage of Hsp90-positive epithelial cells in the COVID-19 and control groups. Magnification 400×, scale bar 20 µm. Data are presented as the means ± SD. *****: *p* < 0.0001.

**Figure 3 cells-10-03136-f003:**
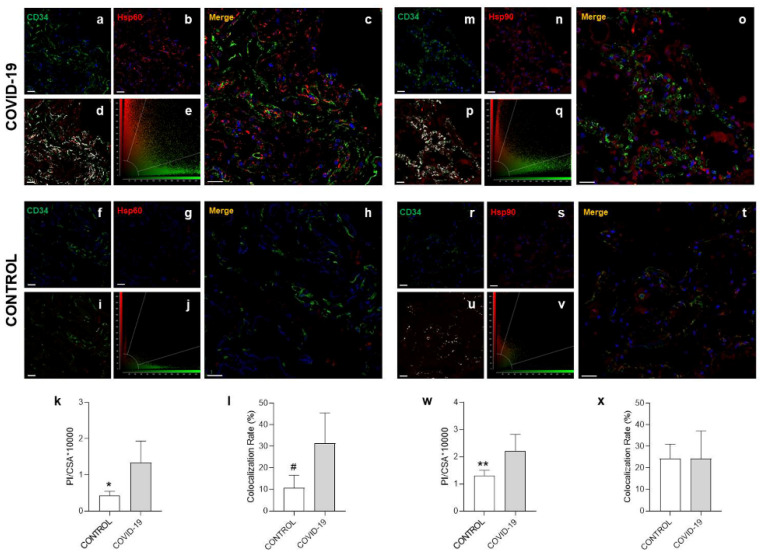
Confocal microscopy analysis demonstrated that Hsp60 and Hsp90 tissue levels increased in the COVID-19 group compared to the control lungs. (**a**–**e**,**m**–**q**) Representative images of COVID-19 lung parenchyma. (**a**,**b**,**m**,**n**) Representative immunofluorescence for CD34 ((**a**,**m**) green), Hsp60 ((**b**), red), and Hsp90 ((**n**), red); nuclei stained with DAPI are shown in blue. (**c**,**o**) The panels show the merged confocal laser scanning image of CD34/Hsp60 (**c**), CD34/Hsp90 (**o**), and nuclei. (**d**,**p**) Merged images show colocalization between CD34 and Hsp60 ((**d**), white) and between CD34 and Hsp90 ((**p**), white). (**e**,**q**) Semiquantification of colocalization parameters between CD34 and Hsp60 (**e**) and between CD34 and Hsp90 (**q**) by a cytofluorogram. (**f**–**j**,**r**–**v**) Representative images of control lung parenchyma. (**f**,**g**,**r**,**s**) Representative immunofluorescence for CD34 ((**f**,**r**) green), Hsp60 ((**g**), red), and Hsp90 ((**s**), red); nuclei stained with DAPI are shown in blue. (**h**,**t**) The panels show the merged confocal laser scanning image of CD34/Hsp60 (**h**), CD34/Hsp90 (**t**), and nuclei. (**i**,**u**) Merged images show colocalization between CD34 and Hsp60 ((**i**), white) and between CD34 and Hsp90 ((**u**), white). (**j**,**v**) Semiquantification of colocalization parameters between CD34 and Hsp60 (**j**) and between CD34 and Hsp90 (**v**) by a cytofluorogram. (**k**,**w**) The staining intensity for Hsp60 (**k**) and Hsp90 (**w**) (bars) expressed as the mean pixel intensity (PI) normalized to the cross-sectional area (CSA) using the Leica Application Suite Advanced Fluorescence software. (**l**,**x**) The bar graph shows the colocalization ratio (%) between CD34 and Hsp60 (**l**) and between CD34 and Hsp90 (**x**). Open bar, control; shaded bar, COVID-19. Data are presented as the means ± SD. * significantly different from COVID-19 group (*p* = 0.0256). # significantly different from COVID-19 group (*p* = 0.0192). ** significantly different from COVID-19 group (*p* = 0.0280). Scale bar 25 μm.

**Figure 4 cells-10-03136-f004:**
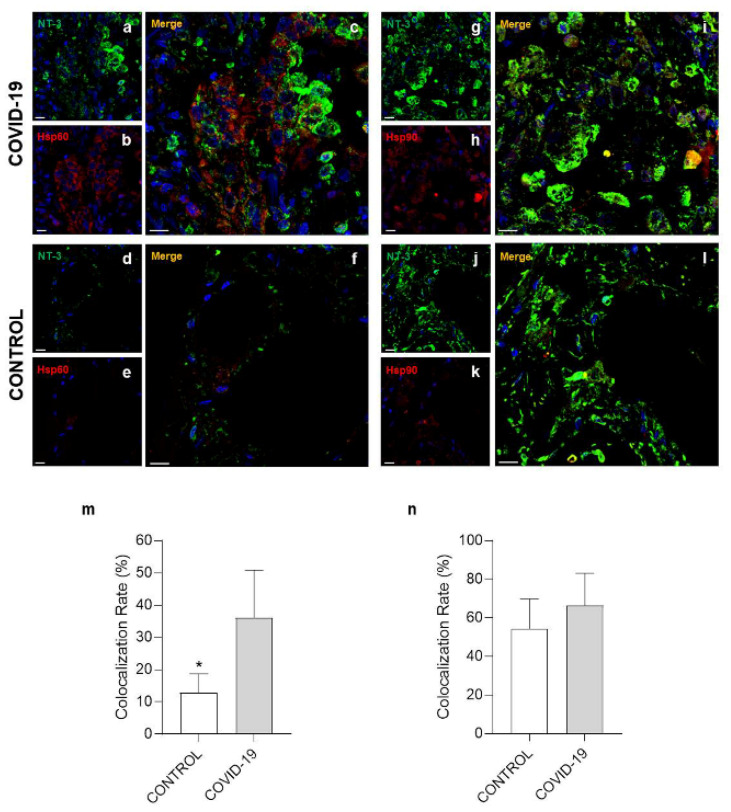
Immunofluorescence for 3-nitrotyrosine, Hsp60 and Hsp90. (**a**–**c**,**g**–**i**) Representative images of COVID-19 lung parenchyma. (**a**,**b**) Representative immunofluorescence for 3-nitrotyrosine ((**a**), green) and Hsp60 ((**b**), red); nuclei stained with DAPI are shown in blue. (**c**) The panel shows the merged confocal laser scanning image of 3-nitrotyrosine, Hsp60, and nuclei. (**g**,**h**) Representative immunofluorescence for 3-nitrotyrosine ((**g**), green) and Hsp90 ((**h**), red); nuclei stained with DAPI are shown in blue. (**i**) The panel shows the merged confocal laser scanning image of 3-nitrotyrosine, Hsp90, and nuclei. (**d**–**f**,**j**–**l**) Representative images of control lung parenchyma. (**d**,**e**) Representative immunofluorescence for 3-nitrotyrosine ((**d**), green) and Hsp60 ((**e**), red); nuclei stained with DAPI are shown in blue. (**f**) The panel shows the merged confocal laser scanning image of 3-nitrotyrosine, Hsp60, and nuclei. (**j**,**k**) Representative immunofluorescence for 3-nitrotyrosine ((**j**), green) and Hsp90 ((**k**), red); nuclei stained with DAPI are shown in blue. (**l**) The panel shows the merged confocal laser scanning image of 3-nitrotyrosine, Hsp90, and nuclei. (**m**,**n**) The bar graph shows the colocalization ratio (%) for 3-nitrotyrosine/Hsp60 (**m**) and 3-nitrotyrosine/Hsp90 (**n**). Open bar, control; shaded bar, COVID-19. Data are presented as the means ± SD. * significantly different from COVID-19 group (*p* = 0.0137). Scale bar 25 μm.

**Figure 5 cells-10-03136-f005:**
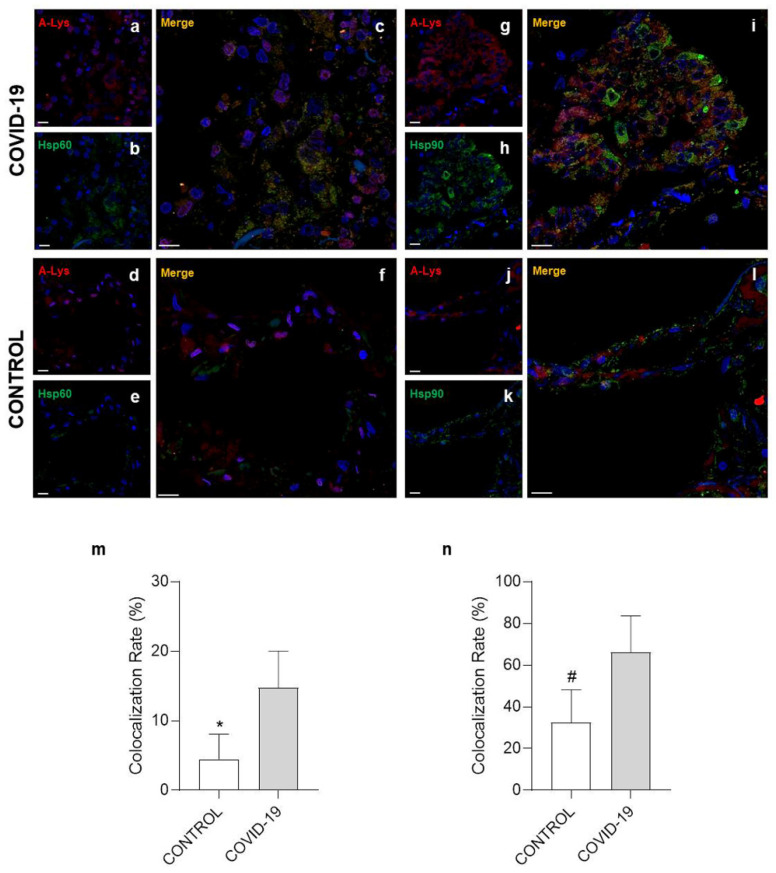
Immunofluorescence for acetylate-lisine, Hsp60, and Hsp90. (**a**–**c**,**g**–**i**) Representative images of COVID-19 lung parenchyma. (**a**,**b**) Representative immunofluorescence for acetylate-lisine ((**a**), red) and Hsp60 ((**b**), green); nuclei stained with DAPI are shown in blue. (**c**) The panel shows the merged confocal laser scanning image of acetylate-lisine, Hsp60, and nuclei. (**g**,**h**) Representative immunofluorescence for acetylate-lisine ((**g**), red) and Hsp90 ((**h**), green); nuclei stained with DAPI are shown in blue. (**i**) The panel shows the merged confocal laser scanning image of acetylate-lisine, Hsp90, and nuclei. (**d**–**f**,**j**–**l**) Representative images of control lung parenchyma. (**d**,**e**) Representative immunofluorescence for acetylate-lisine ((**d**), red) and Hsp60 ((**e**), green); nuclei stained with DAPI are shown in blue. (**f**) The panel shows the merged confocal laser scanning image of acetylate-lisine, Hsp60, and nuclei. (**j**,**k**) Representative immunofluorescence for acetylate-lisine ((**j**), red) and Hsp90 ((**k**), green); nuclei stained with DAPI are shown in blue. (**l**) The panel shows the merged confocal laser scanning image of acetylate-lisine, Hsp90, and nuclei. (**m**,**n**) The bar graph shows the colocalization ratio (%) for acetylate-lisine /Hsp60 (**m**) and acetylate-lisine /Hsp90 (**n**). Open bar, control; shaded bar, COVID-19. Data are presented as the means ± SD. * significantly different from COVID-19 group (*p* = 0.0185). # significantly different from COVID-19 group (*p* = 0.0282). Scale bar 25 μm.

**Figure 6 cells-10-03136-f006:**
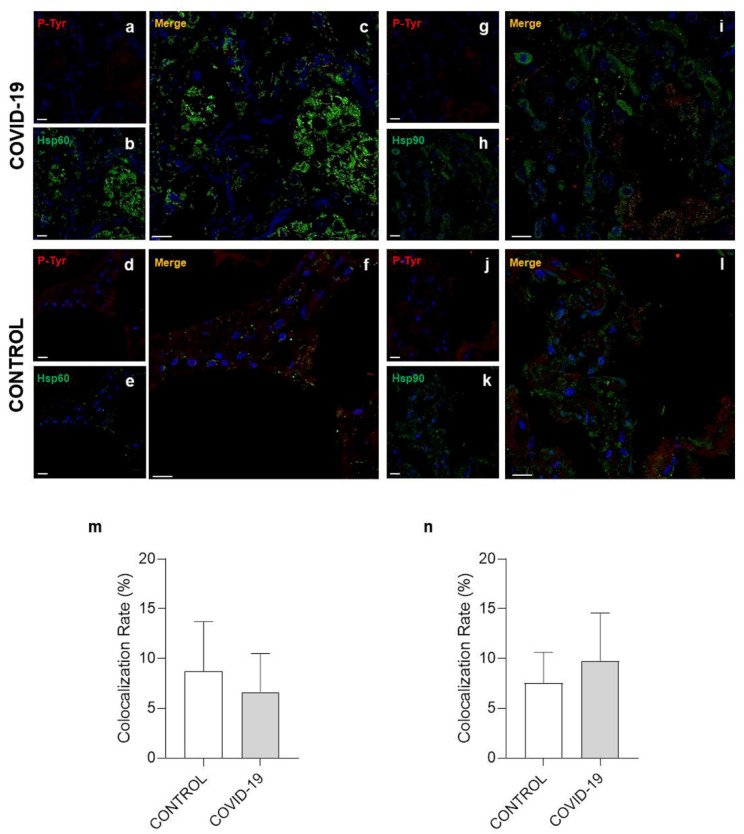
Immunofluorescence for phosphotyrosine, Hsp60, and Hsp90. (**a**–**c**,**g**–**i**) Representative images of COVID-19 lung parenchyma. (**a**,**b**) Representative immunofluorescence for phosphotyrosine ((**a**), red) and Hsp60 ((**b**), green); nuclei stained with DAPI are shown in blue. (**c**) The panel shows the merged confocal laser scanning image of phosphotyrosine, Hsp60, and nuclei. (**g**,**h**) Representative immunofluorescence for phosphotyrosine ((**g**), red) and Hsp90 ((**h**), green); nuclei stained with DAPI are shown in blue. (**i**) The panel shows the merged confocal laser scanning image of phosphotyrosine, Hsp90, and nuclei. (**d**–**f**,**j**–**l**) Representative images of control lung parenchyma. (**d**,**e**) Representative immunofluorescence for phosphotyrosine ((**d**), red) and Hsp60 ((**e**), green); nuclei stained with DAPI are shown in blue. (**f**) The panel shows the merged confocal laser scanning image of acetylate-lisine, Hsp60, and nuclei. (**j**,**k**) Representative immunofluorescence for phosphotyrosine ((**j**), red) and Hsp90 ((**k**), green); nuclei stained with DAPI are shown in blue. (**l**) The panel shows the merged confocal laser scanning image of phosphotyrosine, Hsp90, and nuclei. (**m**,**n**) The bar graph shows the colocalization ratio (%) for phosphotyrosine/Hsp60 (**m**) and phosphotyrosine/Hsp90 (**n**). Open bar, control; shaded bar, COVID-19. Data are presented as the means ± SD. Scale bar 25 μm.

**Table 1 cells-10-03136-t001:** Summary of clinical information of subjects.

**COVID-19**							
**Case**	**Age**	**Sex**	**Days in Hospital**	**Thorax Radiological Findings**	**Clinical Personal History**	**Intubation**	**Initial Clinical Presentation**
1	61	F	23	Ground glass opacities	Hypertension, major depressive disorder	Yes	Confusional state, anemia
2	79	F	19	Ground glass opacities	Cardiovascular disease, COPD, chronic kidney disease	Yes	Fever, anemia, dyspnea/tachypnea
3	72	F	10		Hypertension, cardiovascular disease, COPD, smoker, diabetes mellitus, dyslipidemia, hypothyroidism	Yes	Cardiac arrest
4	68	F	2	Ground glass opacities	Diabetes mellitus, rheumatoid arthritis, polymyalgia	Yes	Fever, dyspnea/tachypnea, diarrhea, anemia
5	70	M	1, Before admission in hospital for emergency, medical consultation.			No	Fever; asthenia, respiratory failure, disorientation
6	42	M	0, Medical consultation, no hospitalization		Alcoholism, pancreatitis	No	Fever, asthenia, respiratory failure, disorientation
**Control**							
**Case**	**Age**	**Sex**	**Cause of Death**				
1	46	F	Death in a road accident				
2	68	M	Death by firearm				
3	66	F	Death from brain hemorrhage				
4	54	F	Death by suicide				
5	70	M	Death by suicide				
6	72	M	Death in a road accident				

**Table 2 cells-10-03136-t002:** Primary antibody used for IHC and IF.

Method	Antigen	Type and Source	Catalog Number	Supplier	Dilution
IHC	CKAE1AE3	Mouse monoclonal	CM 011 A,B,C	BIOCARE medical	1:100
IHC	CK7	Mouse monoclonal	CM 061 A,B,C	BIOCARE medical	1:100
IHC/ IF	CD34	Mouse monoclonal	CM 084 A,B,C	BIOCARE medical	1:100/ 1:50
IHC	CD61	Mouse monoclonal	ACI 3139 A,C	BIOCARE medical	1:100
IHC	CD68	Mouse monoclonal	ACI 3139 A,C	BIOCARE medical	1:100
IHC	Ki67	Mouse monoclonal	API 3156 AA, H	BIOCARE medical	1:100
IHC/ IF	Hsp60	Rabbit polyclonal	sc-13966	Santa Cruz Biotechnology	1:300/ 1:50
IHC/ IF	Hsp90	Mouse monoclonal	sc-59577	Santa Cruz Biotechnology	1:100/ 1:50
IF	Hsp60	Mouse monoclonal	ab13532	Abcam	1:50
IF	Hsp90	Rabbit polyclonal	ab13495	Abcam	1:50
IF	Acetylate-lisine	Rabbit polyclonal	#9441	Cell Signaling Technology	1:50
IF	3-nitrotyrosine	Mouse monoclonal	ab61392	Abcam	1:50
IF	Phosphotyrosine	Rabbit polyclonal	ab179530	Abcam	1:50

Abbreviations: IHC, immunohistochemistry; IF, immunofluorescence.

## Data Availability

The data presented in this study are available on request from the corresponding author.

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
