# Peer review of "Morphological Alterations and Stress Protein Variations in Lung Biopsies Obtained from Autopsies of COVID-19 Subjects"

_cells, 2021, doi:10.3390/cells10113136_

Round 1

Reviewer 1 Report

In this study, it was achieved a histological and an immunomorphological evaluation on lung samples from subjects succumbed to COVID-19 and subjects dead for other causes. The results show that, it was expected, the Hsp60 and Hsp90 immunopositivity was significantly higher in the COVID-19 group compared to the controls. Also, images show that the immunolocalization was in the plasma membrane of the endothelial cells of the COVID-19 subjects.

This is a useful number of observations. My only concern is that “quantifications” are indeed semi-quantifications based on the fluorescence intensity. Given the fact that the authors have at hand enough biological sample, this could have been done in tissue extracts using a more accurate a reliable method, for example, ELISA. Nonetheless, data are valid and can be accepted.

The article is well written, the methodology is presented correctly, and conclusions are supported by the experimental data.

Author Response

Reviewer 1 comment: In this study, it was achieved a histological and an immunomorphological evaluation on lung samples from subjects succumbed to COVID-19 and subjects dead for other causes. The results show that, it was expected, the Hsp60 and Hsp90 immunopositivity was significantly higher in the COVID-19 group compared to the controls. Also, images show that the immunolocalization was in the plasma membrane of the endothelial cells of the COVID-19 subjects. This is a useful number of observations. My only concern is that quantifications are indeed semi-quantifications based on the fluorescence intensity. Given the fact that the authors have at hand enough biological sample, this could have been done in tissue extracts using a more accurate a reliable method, for example, ELISA. Nonetheless, data are valid and can be accepted. The article is well written, the methodology is presented correctly, and conclusions are supported by the experimental data..

Authors’ Reply: Thank you very much for the positive comments regarding our manuscript and the methodology used. We are aware that ELISA or western blot are accurate methodologies for protein quantification. In all honesty, during the experimental steps we tried to obtain tissue extracts from the paraffin blocks using different methods (Qproteome FFPE tissue kit, Qiagen and an in-house protocol) but unfortunately we have not obtained a good quality of the extracts (i.e. poor protein content). For this reason, in this paper we are presenting a morphological analysis with the semi-quantifications based on the fluorescence intensity. For the benefit of clarity, we changed “quantifications”to  semi-quantifications in the text.

Reviewer 2 Report

There are studies demonstrating that one of the contributing factors to severe COVID-19 is autoimmune reaction, caused by immuno-response to SARS-Co-2 virus proteins that bear epitopes resembling human proteins, two of which are Hsp60 and Hsp90. In the studies of this manuscript, Barone R et al investigated the morphological and molecular changes in autopsied lung samples of COVID-19 patients, aiming to discover histological evidence on the involvement of Hsp60 and Hsp90 in endothelialitis and thromboembolisms. They had a total of 12 samples, which were equally divided into 2 groups – COVID-19 and control, the latter of which was comprised of subjects died from causes other than COVID-19. The technique was microscopy. The histological changes in the COVID-19 samples were obvious. With the immunohistochemical changes, while the claimed findings are very interesting – increased positivity of Hsp60 and Hsp90, their membrane localization in endothelial cells, and elevated post-translational modifications – the quality of the images supporting these claims were poor. For example, the confocal images in Fig. 3 showing merely some dots but no cells, making it impossible to judge the meaningfulness of the signals.  Also, most of the immunofluorescent staining in Fig. 4, 5, and 6 for Hsp60, Hsp90, and the PTMs didn’t align with the DAPI staining, raising the concern about non-specific staining. These major issues/concerns need to be resolved before the manuscript can be considered for publication.

Author Response

Reviewer 2 comment: There are studies demonstrating that one of the contributing factors to severe COVID-19 is autoimmune reaction, caused by immuno-response to SARS-Co-2 virus proteins that bear epitopes resembling human proteins, two of which are Hsp60 and Hsp90. In the studies of this manuscript, Barone R et al investigated the morphological and molecular changes in autopsied lung samples of COVID-19 patients, aiming to discover histological evidence on the involvement of Hsp60 and Hsp90 in endothelialitis and thromboembolisms. They had a total of 12 samples, which were equally divided into 2 groups COVID-19 and control, the latter of which was comprised of subjects died from causes other than COVID-19. The technique was microscopy. The histological changes in the COVID-19 samples were obvious. With the immunohistochemical changes, while the claimed findings are very interesting increased positivity of Hsp60 and Hsp90, their membrane localization in endothelial cells, and elevated post-translational modifications the quality of the images supporting these claims were poor. For example, the confocal images in Fig. 3 showing merely some dots but no cells, making it impossible to judge the meaningfulness of the signals. Also, most of the immunofluorescent staining in Fig. 4, 5, and 6 for Hsp60, Hsp90, and the PTMs didnt align with the DAPI staining, raising the concern about non-specific staining. These major issues/concerns need to be resolved before the manuscript can be considered for publication.

Authors’ Reply: Thank you for your very appropriate comment that give us the opportunity to improve the quality of the images. We increased and ameliorated the quality of the DAPI, Hsp60, Hsp90, and the PTMs signals. However, some nuclei are not yet visible, probably due to the orientation of the section cutting or the presence of cellular debris. Please find the new version of the figures attached to the manuscript. We hope that now they will be more explicative and supportive of our claims.

Round 2

Reviewer 2 Report

While more supportive data for the conclusions on the post-translational modifications of Hsp60 and Hsp90 would be optimal, this reviewer found these claims to be provocative and worth to be made public.